Direct imaging of APP proteolysis in living cells

Parenti Niccoló 1 2
Del Grosso Ambra 3
Antoni Claudia 1
Cecchini Marco 3
Corradetti Renato 2
Pavone Francesco S. 1 4
Calamai Martino calamai@lens.unifi.it 1 4
1 European Laboratory for Non-linear Spectroscopy (LENS), University of Florence , Sesto Fiorentino, Florence , Italy
2 Department of Neuroscience, Psychology, Drug Research and Child Health, University of Florence , Florence , Italy
3 NEST, Istituto Nanoscienze-CNR and Scuola Normale Superiore , Pisa , Italy
4 National Institute of Optics, National Research Council of Italy (CNR) , Florence , Italy
Haraguchi Tokuko
Electronic publication date: 2017 Apr 12
Publication date: 2017
Volume: 5
Electronic Location ID: e3086
Received 2016 Oct 6; Accepted 2017 Feb 13
Copyright: ©2017 Parenti et al.
Copyright year: 2017
Copyright holder: Parenti et al.
License: This is an open access article distributed under the terms of the Creative Commons Attribution License, which permits unrestricted use, distribution, reproduction and adaptation in any medium and for any purpose provided that it is properly attributed. For attribution, the original author(s), title, publication source (PeerJ) and either DOI or URL of the article must be cited.
License URL: https://creativecommons.org/licenses/by/4.0/

Keywords: Alzheimer’s disease, Amyloid precursor protein, Bace1, Imaging, Facs, Fluorescence microscopy, In vivo assay, Adam10

Funding: European Union’s Horizon 2020 research and innovation programme EU-H2020 654148 Italian Ministry of Health in the framework of the project GR-2011-02349626 Italian Ministry for Education, University and Research in the framework of the Flagship Project NANOMAX The research leading to these results has received funding from the European Union’s Horizon 2020 research and innovation programme under grant agreement no 654148 Laserlab-Europe (EU-H2020 654148), from the Italian Ministry of Health in the framework of the project “Automated digital scanning and diagnosis of tissues using multimodal non-linear optical microscopy” (GR-2011-02349626), and from the Italian Ministry for Education, University and Research in the framework of the Flagship Project NANOMAX. The funders had no role in study design, data collection and analysis, decision to publish, or preparation of the manuscript.

==============================
Alzheimer’s disease is a multifactorial disorder caused by the interaction of genetic, epigenetic and environmental factors. The formation of cytotoxic oligomers consisting of Aβ peptide is widely accepted as being one of the main key events triggering the development of Alzheimer’s disease. Aβ peptide production results from the specific proteolytic processing of the amyloid precursor protein (APP). Deciphering the factors governing the activity of the secretases responsible for the cleavage of APP is still a critical issue. Kits available commercially measure the enzymatic activity of the secretases from cells lysates, in vitro. By contrast, we have developed a prototypal rapid bioassay that provides visible information on the proteolytic processing of APP directly in living cells. APP was fused to a monomeric variant of the green fluorescent protein and a monomeric variant of the red fluorescent protein at the C-terminal and N-terminal (mChAPPmGFP), respectively. Changes in the proteolytic processing rate in transfected human neuroblastoma and rat neuronal cells were imaged with confocal microscopy as changes in the red/green fluorescence intensity ratio. The significant decrease in the mean red/green ratio observed in cells over-expressing the β-secretase BACE1, or the α-secretase ADAM10, fused to a monomeric blue fluorescent protein confirms that the proteolytic site is still accessible. Specific siRNA was used to evaluate the contribution of endogenous BACE1. Interestingly, we found that the degree of proteolytic processing of APP is not completely homogeneous within the same single cell, and that there is a high degree of variability between cells of the same type. We were also able to follow with a fluorescence spectrometer the changes in the red emission intensity of the extracellular medium when BACE1 was overexpressed. This represents a complementary approach to fluorescence microscopy for rapidly detecting changes in the proteolytic processing of APP in real time. In order to allow the discrimination between the α- and the β-secretase activity, we have created a variant of mChAPPmGFP with a mutation that inhibits the α-secretase cleavage without perturbing the β-secretase processing. Moreover, we obtained a quantitatively robust estimate of the changes in the red/green ratio for the above conditions by using a flow cytometer able to simultaneously excite and measure the red and green fluorescence. Our novel approach lay the foundation for a bioassay suitable to study the effect of drugs or particular conditions, to investigate in an unbiased way the the proteolytic processing of APP in single living cells in order, and to elucidate the causes of the variability and the factors driving the processing of APP.

Introduction

In the last decades the incidence of Alzheimer’s disease (AD) has risen due to an increase in the lifespan, representing more and more a serious social problem. The first steps in the pathological cascade involve protein quality control impairment and the failure of misfolded proteins clearance mechanisms, with subsequent accumulation and oligomeritazion of neurotoxic amyloid-β peptide. Aβ oligomers decrease the synaptic efficacy, activate microglia and astrocytes causing an inflammatory response, alter the degree of tau’s phosphorylation leading to tangles, deposit in the brain blood vessels and cause neuronal loss and neurotransmitter deficits (Selkoe & Hardy, 2016). The generation of amyloid-β peptide results from the sequential cleavage of the amyloid precursor protein (APP) by β-secretase and γ-secretase (Haass, 2004; De Strooper, 2010). This process is in competition with an alternative non-pathogenic pathway lead by the α-secretase (De Strooper, 2010). APP is a plasma membrane glycoprotein whose original function remains unclear (Senechal, Larmet & Dev, 2006). The β-secretase and the other secretases have become obvious therapeutic targets for drug research in AD (MacLeod et al., 2015). Several strategies have mainly focused on the inhibition of β-secretase and γ-secretase or, alternatively, on Aβ clearance. Unfortunately, none of these pharmacological therapies has yet passed the phase II/III trial due to a lack of encouraging results, or to the onset of adverse side effects (MacLeod et al., 2015). Despite the recent failures, significant hope still lies around the secretase targeting approach. With this regard, the full understanding of the factors governing the rate of proteolytic processing of APP is imperative. Commercial kits (Sigma-Aldrich, Abcam) to evaluate the activity of the β-secretase BACE1 from human and animal samples are available. These assays use a specific BACE1 substrate conjugated to two reporter molecules. In the uncleaved form, the fluorescence of the reporter is quenched by the physical proximity of the other molecule. After the cleavage by β-secretase, the two molecules are separated and a fluorescent signal can be acquired with intensity proportional to the enzymatic activity. This type of kit is very efficient in detecting the activity of BACE1 from purified samples, but remains an in vitro evaluation. By contrast, we have developed a consistent bioassay to quantify in an unbiased way the the proteolytic processing of APP in single living cells. The key element of our assay is a chimeric construct of APP with two different fluorescent proteins fused at the opposite terminals. Changes in the proteolytic processing of APP are seen as changes in the intensity ratio of the two fluorescent protein tags. We demonstrate that our mChAPPmGFP construct is efficiently recognized and cleaved by both α- and β-secretase. The activity of each individual secretase can be discriminated by using specific siRNAs or, alternatively, by introducing specific mutations at the cleavage sites of APP. Here, we have created a mutated variant of mChAPPmGFP that can be cleaved almost exclusively by β-secretase (mChAPPP1mGFP). The mutation, a Lys to Val substitution at position 612 of APP695, just before the α-secretase cleavage site (this position is called P1), has already been used to verify the sequence specificity of α-secretase cleavage (Sisodia, 1992). Sisodia tested a series of mutants at the P1 position, and found by immunoprecipitation studies that only one, the Lys to Val substitution, prevents the α-secretase cleavage.

Finally, we obtained a quantitatively robust validation of our bioassay by using a flow cytometer able to simultaneously excite and measure the emission intensity of the two fluorescent proteins.

Materials and Methods

Cloning

To generate the fusion constructs of mChAPPmGFP and mChAPP770mGFP we used APP695 and APP770 contained in the vector pCMV-AC-GFP (Origene, Rockville MD, USA). First, we inserted new restriction sites (NheI, AscI) after the signal peptide using the GeneArt Site-Directed Mutagenesis Kit (ThermoFisher Scientific), using GCCGCCTGGACGGCTCGGGCGGCTAGCCAGGGCGCGCCTCTGGAGGTACCCACTGATGG as forward primer, and CCATCAGTGGGTACCTCCAGAGGCGCGCCCTGGCTAGCCGCCCGAGCCGTCCAGGCGGC as reverse primer. The mCherry sequence was amplified from its original plasmid (pmCherry-C1 vector, Clontech) with a PCR reaction (forward primer: TAAGCAGCTAGCATGGTGAGCAAGGGCGAGGAG, reverse primer TGGCG CGCCTGTTCCACGACTCTTGTACAGCTCGTCCATG) adding the restriction sequences for the enzymes NheI and AscI. After digestion, the sequence of mCherry was inserted at the N-term of APP. In order to substitute the dimerizing TurboGFP present in the pCMV-AC-GFP vector with the monomeric green fluorescent protein mTagGFP2, mCherry-APP695 and mCherry-APP770 were excised from their original plasmid using the restriction enzymes SgfI and NotI and inserted in the plasmid pCMV6-AN-mGFP (commercial name of mTagGFP2, Origene) to obtain the final fusion constructs mCherry-APP695-mGFP and mCherry-APP770-mGFP. The QuikChange Lightning Site-Directed Mutagenesis Kit (Agilent) was used to mutagenize the mChAPPmGFP construct in order to create a new plasmid (mChAPPP1mGFP) carrying the P1 mutation, a Lys to Val substitution at P1 position (amino acid 612 of APP695), that is the amino acid located before the α-secretase cleavage site (Sisodia, 1992) (forward primer: GAAGTTCATCATCAAGTATTGGTGTTCTTTGC, reverse primer: GCAAAGAACACCAATACTTGATGATGAACTTC).

To generate the fusion construct Bace1-mBFP and HA-Bace1-mBFP, we started from Bace1 contained in the vector pCMV6-ENTRY (NM_012104, Origene, Rockville, MD, USA). We opened the vector using the restriction enzymes NotI and SacII. mBFP (original name mTagBFP2) was extracted with a PCR reaction from the plasmid pmTagBFP2-N1 (a gift from Vladislav Verkhusha—Addgene plasmid # 34633) (Subach et al., 2011) (forward primer: TAAGCAGCGGCCGCGAATGAGCGAGCTGATTAAGG, reverse primer: CCGAGTCCGCGGTTAATTAAGCTTGTGCCCCAG) to add the restriction sequences of the enzymes NotI and SacII. After digestion, the sequence of mBFP was inserted in the plasmid of Bace1 previously opened with the same enzymes, to obtain the final fusion construct Bace1-mBFP. With a site-directed mutagenesis reaction (ThermoFisher Scientific) we added the restriction sites for NheI and AscI at the N-term of Bace1-mBFP (forward primer: GGAGTGCTGCCTGCCCACGGCGCTAGCCAGGGC GCGCCtACCCAGCACGGCATCCGGCTG, reverse primer CAGCCGGATGCCGTGCTGGGTAGGCGCGCCCTGGCTAGCGCCGTGGGCAGGCAGCACTCC). We opened the plasmid of Bace1-mBFP and inserted the annealed oligos containing the sequence for the HA tag (forward oligo: CTAGCTATCCGTACGACGTACCAGACTACGCAGG, reverse oligo: CGCGCCTGCGTAGTCTGGTACGTCGTACGGATAG). After ligation we obtained the plasmid of the construct HA-Bace1-mBFP.

ADAM10mBFP was created by the fusion of the mTagBFP2 gene downstream of the ADAM10 gene contained in the pRK5M-ADAM10 construct (a gift from Rik Derynck—Addgene plasmid # 31717) (Liu et al., 2009). The final part of the sequence coding for ADAM10 was mutagenized in order to introduce two new restriction sites, unique within the entire sequence of the plasmid. The site-directed mutagenesis was performed with the QuikChange Lightning Site-Directed Mutagenesis Kit (Agilent Technologies), using GAGGACCTGCTGCGTACGGTCGCGCGCGCTTGGCCGCCA as forward primer and TGGCGGCCAAGCGCGCGCGACCGTACGCAGCAGGTCCTC as reverse primer, containing the restriction sequences for the BsiWI e BssHII enzymes. The mTagBFP2 sequence was amplified from its original plasmid (pmTagBFP2-N1, Addgene) with a PCR reaction (forward primer GAGCGTACGATGAGCGAGCTGATTAAGGAG, reverse primer GTAGCGCGCTTAATTAAGCTTGTGCCCCAG, the forward primer containing the BsiWI restriction site and the reverse the BssHII one). After digestion, the sequence of BFP2 was inserted at the C-terminal of ADAM10.

apAPPha was created from mChAPPmGFP using the restriction sites for NheI and AscI at the N-term. First, we opened the plasmid with these enzymes and inserted the annealed oligos containing the sequence for the genetic tag AP (forward oligo: CTAGCGAGGGCCTGAACGATATCTTCGAGGCCCAGAAGATCGAGTGGCACGAGAGTGG, reverse oligo: CGCGCCACTCTCGTGCCACTCGATCTTCTGGGCCTCGAAGATATCGTTCAGGCCCTCG). Afterwards, we opened the plasmid with the restriction enzimes NotI and SacII in order to substitute mGFP with the annealed oligos containing the sequence for the genetic tag HA (forward oligo: GGCCGCTCGAGTATCCGTACGACGTACCAGACTAC GCAGTTTAAACCCGC, reverse oligo: GGGTTTAAACTGCGTAGTCTGGTACGTCGTACGGATACTCGAGC).

Cell cultures and transfection

Human SH-SY5Y neuroblastoma cells (A.T.C.C. Manassas, VA, USA) were cultured in Dulbecco’s Modified Eagle’s Medium (DMEM) (ThermoFisher Scientific—Waltham, Massachusetts, USA) F-12 supplemented with 10% FBS, 1% penicillin/streptomycin solution. HEK cells were cultured in DMEM Glutamax supplemented with 10% FBS, 1% penicillin/streptomycin solution. Primary rat hippocampal neurons were obtained after dissociation of hippocampi purchased from Brainbits, UK. Neurons were plated on coverslips coated with poly-L-ornithine and grown in Neurobasal medium supplemented with 2% B27, 0.5 mM glutamine, 12.5 mM glutamate and .01% penicillin/streptomycin. All the cell types were cultured in a humidified chamber at 5% CO2 and 37 °C. Neuroblastoma SH-SY5Y cells were transfected with Fugene HD (Promega, USA) 24–48 h before imaging. HEK cells were transfected with Lipofectamine 2000 (ThermoFisher Scientific) 24–48 h before imaging. To transfect hippocampal neurons DNA was mixed with Lipofectamine 2000 in 0.5 ml of Neurobasal supplemented with 0.5 mM glutamine at 37 °C in 5% CO2 for 60 min. Afterwards neurons were rinsed with Neurobasal, transferred to the original medium at 37 °C 5% CO2 and imaged after 24–48 h. apAPPha was co-transected with BirA, a biotin ligase that specifically biotinylates the AP tag. Biotin was added to the cell medium (final concentration 10 µM) 8 h after transfection and 12 h before imaging (Howarth et al., 2005). siRNA Silencer® Validated siRNA against BACE1 (105154) and Silencer® Negative Control siRNA were purchased from ThermoFisher. Figure S5 shows the silencing efficiency of Bace1-siRNA.

Cell imaging

Live cell imaging was performed on a Nikon Eclipse TE300 C2 LSCM (Nikon, Japan) equipped with a Nikon 60×  immersion oil objective (Apo Plan, NA 1.4). Cells were growth on 18 mm coverslips and mounted on a custom chamber containing 1 ml of Leibovitz’s medium (ThermoFisher). For immunolabelling, cells were fixed with 4% PFA, rinsed with PBS (+MgCl2 0,5 mM, +CaCl2 0,8 mM) and permeabilized with 0,075% Triton X. After rinsing with PBS and blocking with 4% BSA PBS, cells were labelled with primary (1:1,000 rabbit anti-HA ab9110 Abcam, UK) and secondary antibodies (goat anti-rabbit Alexa 488, ThermoFisher, diluted at 1:500), or streptavidin Alexa 568 (ThermoFisher diluted at 1:500), diluted in PBS with 4% BSA. After rinsing again with PBS, coverslips were mounted on a glass slide and imaged with LSCM, equipped with Coherent CUBE (diode 405 nm), Melles Griot (Argon 488 nm) and Coherent Sapphire (Sapphire 561 nm) lasers. Emission filters for imaging were 452/45 nm, 514/30 nm and 595/60 nm. For surface labeling, living SH-SY5Y cells were growth on 18 mm coverslips, rinsed with PBS and incubated with primary antibodies (1:100 rabbit anti-mCherry ab167453, or 1:1000 rabbit anti-HA ab9110 Abcam, UK) diluted in PBS with 4% BSA for 30 min on ice, to inhibit endocytic events. After rinsing with PBS, cells were incubated with secondary antibodies (1:500 goat anti-rabbit Alexa 405) diluted in PBS with 4% BSA for 15 min. After rinsing again the coverslips, cells were mounted on a custom chamber filled with Leibovitz’s medium and imaged with the same LSCM configuration mentioned above.

Western blotting

Human SH-SY5Y neuroblastoma cells were cultured in 6 wells plates. 24 h after transfection, the extracellular medium was recovered and the cells were lysed using 300 µl of PBS and Laemmli buffer. Lysed cells were collected in safe lock 1.5 ml tubes and immersed in boiling water for 5 min, then sonicated two times for 15 s and boiled again for 5 min. The recovered medium was centrifuged at 10,000 rpm for 5 min in order to remove cell debrees and other impurity, and then treated as the cell lysate. Samples were loaded in pre-casted polyacrylamide gels (ThermoFisher Scientific, MA, USA) and blotted on PVDF membranes (ThermoFisher Sicnetific, MA, USA). Membranes were blocked in 5% milk in PBST overnight at 4 °C or 1 h at room temperature. After blocking, membranes were rinsed three times with PBST and incubated at room temperature with primary antibody (in PBST with 4% BSA) for 1 h and for an additional hour with secondary anti-rabbit antibody conjugated to horseradish peroxidase (1:20000). The primary antibodies were: rabbit anti-mCherry (1:1000, Abcam, UK, ab167453), rabbit anti-APP (1:200, Abcam ab15272), rabbit anti-vimentin (1:2500, Abcam ab92547 EPR3776), mouse anti-aβ (1:100, Abcam ab11132 DE2b4), mouse anti-GFP (1:1000 Abcam, ab291 LGB-1). Membranes were rinsed with the HRP substrate Pico Dura (ThermoFisher Scientific, MA, USA) and imaged after 5 min with a Chemidoc MP system (Bio-Rad, CA, USA). In order to perform the stripping procedure, the blotted PVDF membranes were covered with Seppro Stripping Buffer (Sigma Aldrich, MO, USA) and incubated for 5–10 min at room temperature. The stripping buffer was then discarded and the incubation was repeated with fresh Seppro buffer for 5–10 min. After discarding the buffer, membranes were washed for two times with PBS for 10 min, then washed for two times with TBST for 5 min and blocked.

Fluorescence spectroscopy

Fluorescence spectra from 585 nm to 700 nm were acquired at 25 °C in a 10 × 10 mm quartz cuvette using a Perkin-Elmer LS 55 spectrofluorimeter (Waltham, MA, USA) equipped with a thermostated cell holder attached to a Haake F8 water bath (Karlsruhe, Germany). Excitation wavelength was 550 nm.

FACS

Cells were harvested after mild trypsin treatment and centrifuged for 5 min at 330 g; the obtained pellets were washed and re-suspended in PBS. We cannot exclude that the treatment with trypsin might affect the integrity of the ectodomain of mChAPPmGFP; however, all the conditions of transfection tested would have been affected equally without influencing the conclusion of our experiments. Flow cytometry was performed by using a S3 flow cytometer (Bio-Rad) equipped with 488 and 561 nm diode-pumped solid-state lasers. mGFP was excited using the 488 nm laser, and the emitted fluorescence was collected through a 586/25 nm band-pass filter; mCherry was excited with the 561 nm laser and its fluorescence was collected through a 615/25 nm band-pass filter. Data were analysed using ProSort software (Bio-Rad), by dividing the scatter plot of mGFP versus mCherry into two equally populated regions under control conditions. The cellular debris was excluded from the quantification by gating cells on the Forward scatter area/Side scatter area (FSC/SSC) graph, while doublets were excluded by gating the cells on the Forward scatter height/Forward scatter area (FSC/FSC) graph; at least 10,000 events were acquired for each sample.

Results

mChAPPmGFP is correctly targeted to the plasma membrane

APP695, the isoform mainly expressed in the central nervous system (Nalivaeva & Turner, 2013), was fused to a monomeric variant of the green fluorescent protein (mGFP) at the C-terminal and to a monomeric red fluorescent protein (mCherry) at the N-terminal (Fig. 1). In this way the extremities of APP situated at the opposite sides of the plasma membrane are labeled with two different fluorescent proteins. Once the N-term mCherry containing domain is cleaved, the fluorescence intensity ratio between the red and green signals changes. It is thus possible to monitor in living cells a higher or lower processing of APP in function of the ratio shift. From herein, we will refer to this chimeric construct bearing the APP695 isoform as to mChAPPmGFP. The same procedure was repeated to obtain the fusion construct of APP770 (mChAPP770mGFP), an alternative isoform expressed more ubiquitously (Zheng & Koo, 2011).

Figure 1 Schematic representation of the bioassay rationale.

(A) Design of the various chimeric constructs. The N-terminal tags are inserted after the signal peptide (SP) that targets APP to the plasma membrane. HA and AP are two short amino acid sequences. HA is recognized by a specific antibody and AP is biotinylated by a specific enzyme. (B) Once mChAPPmGFP is cleaved by β-secretase, the sAPPβ fragment carrying mCherry (mCherry-sAPPβ) is released into the medium, causing a change in the ratio between the red and green fluorescence intensity.

mCherry was fused immediately after the signal peptide, which is necessary to target APP to the plasma membrane, and away from the site recognized and cleaved by the β-secretase. In order to test if mChAPPmGFP is correctly targeted to the plasma membrane, we performed a surface immunolabelling of transfected human SH-SY5Y neuroblastoma cells with a primary anti-mCherry antibody (Fig. 2A). Confocal laser scanning microscopy (CLSM) imaging reveals a good degree of labeling only in cells expressing mChAPPmGFP, confirming that mCherry is correctly folded and present on the surface of the cell. We observed red clusters not colocalising with anti-mCherry, thus indicating their intracellular localization.

To confirm that the full-length chimeric mChAPPmGFP protein was expressed, we examined the colocalization of mCherry and mGFP in neuroblastoma cells (Figs. 2B–2D) and rat hippocampal neurons (Figs. 2E–2G). The toolbox available with the JACoP plug-in (Bolte & Cordelieres, 2006) under the ImageJ software (Schneider, Rasband & Eliceiri, 2012) was used to quantify the degree of colocalization. We found that in most cells the plot of the pixel intensity of the green against that of the red channel (scatterplot) follows a linear relationship, with an average Pearson’s coefficient (an estimate of the quality of the linear relationship between the two signals with 1 standing for complete positive correlation) of 0.91. We also found an average Manders’ overlap coefficient (Manders et al., 1992) M1 (indicating the fraction of mCherry overlapping with mGFP) of 0.97 and an M2 (fraction of mGFP overlapping mCherry) of 0.85. The measured Costes’ randomization P-value (Costes et al., 2004) of 100% excludes that the co-localization of pixels is due to chance. An additional method such as Li et al. (2004) intensity correlation analysis further supports a very good degree of colocalization, indicating an abundant presence of full-length mChAPPmGFP. We recurrently observed small red fluorescent punctae, at which did not correspond a comparable level of green fluorescence intensity. Similar data were obtained for mChAPP770mGFP (Fig. S1).

Figure 2 Correct targeting of mChAPPmGFP and colocalization of mCherry and mGFP.

(A) Maximum intensity projection of a confocal z-stack of fixed human SH-SY5Y cells transfected with mChAPPmGFP and surface labeled with anti-mCherry coupled to secondary Alexa 405 antibody (blue). Maximum intensity projections of confocal z-stacks of living human SH-SY5Y cells (B) and rat hippocampal neurons (E) transfected with mChAPPmGFP. (H) Maximum intensity projection of a confocal z-stack of fixed and permeabilized SH-SY5Y cells transfected with apAPPha and labeled with streptavidin Alexa 568 (red), which binds to the bionitylated AP tag, and anti-HA coupled to secondary Alexa 488 antibody (green). The high degree of co-localisation of the red and green signals evident from the linear correlation of scatterplots (C, F, I) and the exponential shape of the Li’s intensity correlation analysis (D, G, J) is confirmed by the Pearson’s (Pc) and Manders’coefficients (M1 and M2) close to 1, and by the intensity correlation quotients (ICQ) close to 0.5. The probability of obtaining the observed Pcs by chance is inversely correlated with the Costes’ randomization P value, which is 100% in all cases. Scalebars, 10 µm.

Due to their size and consequent steric hindrance, the two fluorescent tags could alter the distribution features of APP. However, we did not observe any significant difference in the overall cellular distribution after immunolabeling of an overexpressed construct of APP695 bearing two small tags (AP Howarth et al., 2005 and HA Schembri et al., 2007, apAPPha) at the two opposite terminals (Figs. 2H–2J). As previously reported (Collins et al., 2010; Guo et al., 2012; Haass et al., 2012), a large fraction of intracellular mChAPPmGFP is localized at the level of the Golgi apparatus, in the perinuclear region (Fig. S2). In agreement with an elegant study investigating the subcellular localization of endogenous full length APP in neurons (Guo et al., 2012), mChAPPmGFP also shows partial co-localization with the endoplasmic reticulum, but very little with early endosomes and lysosomes (Fig. S2). We find this study of particular relevance since it shows with a knock-out model that many antibodies against APP commonly used for immunofluorescence experiments give rise to non-specific labeling.

Figure 3 Red/green fluorescence intensity ratio and intercellular variability of mChAPPmGFP processing.

Maximum intensity projections of confocal z-stacks of living human SH-SY5Y cells (A) and rat hippocampal neurons (B) transfected with mChAPPmGFP. The average red/green fluorescence ratio can fluctuate within the same population of cells (examples indicated by coloured arrows). The red/green ratio image has been mean filtered (2 pixels) only for representation purposes. Scalebar 10 µm.

Fluorescence intensity ratio heterogeneity

We then measured the ratio between the red and the green fluorescence intensity in neuroblastoma cells (Fig. 3A) and rat hippocampal neurons (Fig. 3B). The ratio image was generated using the Ratio Plus available on ImageJ (Schneider, Rasband & Eliceiri, 2012). Regions of interest (ROIs) for the calculation of the mean ratio value per cell were chosen by using a constant threshold value on the maximum projection of mGFP stacks. The ROIs were then applied to the corresponding ratio image. The gain used for the acquisition of neuronal images was different from that of neuroblastoma cells due to differences in the level of protein expression. We observed a certain degree of intracellular heterogeneity in the red/green ratio, with lower red/green ratio in the peripheral areas of the cells. In addition, we observed a relatively high intercellular variability among the same population of transfected neuronal or neuroblastoma cells.

mChAPPmGFP is cleavable by Bace1

In order to check if our fusion protein was correctly folded, recognized and cleaved, we co-transfected SH-SY5Y (Fig. 4) and neuronal cells (Fig. 5) with mChAPPmGFP together with β-secretase Bace1 fused with mTagBFP2 (mBFP) at the C-term. We found that overexpressed BACE1 is aboundantly present on the plasma membrane (Fig. S3). When Bace1-mBFP cleaves mChAPPmGFP on the cell surface, the soluble fragment mCherry-sAPPβ is released in the extracellular medium. We compared the mean red/green ratio in cells expressing mChAPPmGFP with or without Bace1-mBFP (Figs. 4A–4C and 5). While we did not find statistically significant differences in cells transfected with siRNA against Bace1 (although an unpredicted minor reduction in the ratio is visible), the overexpression of Bace1-mBFP dramatically decreased the ratio. Although APP might undergo a major processing by α-secretase under physiological conditions, the contribution of the latter is negligible when APP is overexpressed with BACE1. Cells transfected with mChAPP770mGFP showed a comparable trend.

Figure 4 Proof of principle: overexpressed β-secretase cleaves efficiently mChAPPmGFP.

(A) Maximum intensity projection of a confocal z-stack of living human SH-SY5Y cells co-transfected with mChAPPmGFP and Bace1-mBFP. The red/green fluorescence ratio image has been mean filtered (2 pixels) only for representation purposes. Scalebar, 10 µm. (B) The different mean red/green fluorescence intensity ratio values of the two cells outlined in ratio image (A) are correlated to the overexpression of Bace1-mBFP (error bars, s.d.). (C) Dot blot of the mean red/green ratio of SH-SY5Y cells under different conditions of transfection. n > 15 for each condition; ***, p < 0.001 according to unpaired Student-t test with unequal variance, run with respect to cells transfected only with mChAPPmGFP or mChAPP770mGFP. (D) Superimposed westernblots of transfected SH-SY5Y cell lysates (magenta) and extracellular media (cyan), labelled with anti-mCherry. The lysate blot was repeatedly stripped and reprobed with anti-GFP, anti-APP and anti-A β. The lower band corresponds to mCherry-sAPP, while the higher to full-length mChAPPmGFP. (E) The extracellular medium of SH-SY5Y cells under different conditions of transfection was analysed with a fluorescence spectrophotometer. Emission spectra were collected following excitation at 550 nm.

Figure 5 mChAPPmGFP is properly processed in neurons.

Maximum intensity projections of confocal z-stacks of living rat hippocampal neurons co-transfected with mChAPPmGFP and mBFP (A) or Bace1-mBFP (B). The red/green ratio images were mean filtered (2 pixels) only for representation purposes. The insets correspond to the magnified regions outlined in the ratio images. Scalebars 10 µm.

Equivalent results were obtained when performing western blots of both cell lysate and extracellular medium of SH-SY5Y cells co-transfected with mChAPPmGFP and Bace1-mBFP or Bace1-siRNA, using anti-mCherry as primary antibody for chemoluminescent detection (Fig. 4D). In the lysate of cells co-transfected with mChAPPmGFP and Bace1-siRNA, two bands were detected, one more intense at 152 KDa, corresponding to full-length mChAPPmGFP (expected molecular weight 134 KDa), and one at 125 KDa, possibly corresponding to mCherry-sAPPα (expected molecular weight 100 KDa) generated by endogenous α-secretase. For cells co-transfected with mChAPPmGFP and Bace1-mBFP, the same two bands were visible, but in this case the lower band was more intense, due to an increased production of mCherry-sAPPβ, which has a molecular weight (expected molecular weight 94 KDa) comparable to mCherry-sAPPα. Blotting of the extracellular medium revealed only one band at 121 KDa, corresponding to the released mCherry-sAPPfragment. A considerably higher intensity band was found in the case of cells co-transfected with Bace1 (increased production of mCherry-sAPPβ) as compared to those co-transfected with Bace1-siRNA (mostly mCherry-sAPPα). Analogous results were obtained for the cell lysate and the extracellular medium of cells transfected with mChAPP770mGFP, with the bands being at higher molecular weights: ∼163 KDa for the uncleaved mChAPP770mGFP (expected molecular weight 142 KDa) and ∼138 KDa for the mCherry-sAPP770β (expected molecular weight 102 KDa).

Membrane stripping and reprobing with anti-GFP, anti-APP (directed against the N-terminal 44–62 amino acids) or anti-Aβ (directed against amino acids 1–17 of Aβ) generated analogous results. At long time exposures of the blot, bands corresponding to the C-terminal fragments of mChAPPmGFP (C83/89-mGFP and AICD-mGFP) are visible (Fig. S4). These data corroborate further that our construct is recognized and processed by BACE1.

Furthermore, we collected and analyzed the supernatant media from different transfection conditions of SH-SY5Y cells with a fluorescence spectrophotometer. In agreement with the above results, we observed higher fluorescence intensity at 610 nm (maximum emission peak for mCherry) in cells co-expressing mChAPPmGFP together with Bace1-mBFP (Fig. 4E). This result further supports that β-secretase recognizes and cleaves our fusion APP construct, which is then in part released in the extracellular medium as mCherry-sAPPβ. Similar results were found in HEK cells (Fig. S5). Our data show an unexpected slight increase in fluorescence intensity at 610 nm in the cells co-transfected with Bace1-siRNA compared to the control condition. Overall these results suggest that the site cleaved by β-secretase is still accessible and that it is possible to follow the processing of APP as function of the ratio shift.

We finally validated our results with FACS flow cytometry, a statistically robust technique capable of analyzing the red and green fluorescence intensity for a large number of single cells. SH-SY5Y cells were co-transfected with mChAPPmGFP and Bace1-mBFP, or control siRNA, or Bace1-siRNA. Two different areas in the red versus green fluorescence scatter plot were arbitrarily chosen so that both areas contained the same number of cells co-transfected with mChAPPmGFP and control siRNA (Fig. 6). The overexpression of Bace1-mBFP changed significantly the ratio between the number of cells in the red and green areas. We observed only a slight, although not significant, increase in the ratio when cells were co-transfected with siRNA directed against Bace1. Comparable results were found using HEK cells (Fig. S6).

Figure 6 FACS flowcitometry as additional way to screen mChAPPmGFP processing.

(A) Each dot in the scatter plot represents the green and red fluorescence intensity of a single cell. The two different areas (red and green) in the scatter plot were arbitrarily chosen so that both areas contained the same number of SH-SY5Y cells co-transfected with mChaPPmGFP and control siRNA. These regions were used to gate the scatter plots of cells co-transfected with mChAPPmGFP and Bace1-mBFP, Bace1-siRNA, or control siRNA, and calculate the mean ratio between the number of cells in the red area over the number of cells in the green area (B). n = 3 independent experiments; error bars, s.d.; p < 0.0001 according to unpaired Student-t test, run with respect to cells transfected with mChAPPmGFP and control siRNA.

Figure 7 The mChAPPP1mGFP mutant is not efficiently cleaved by the α-secretase ADAM10.

(a) Maximum intensity projection of confocal z-stacks of living human SH-SY5Y cells under different conditions of transfection. The red/green ratio images were mean filtered (2 pixels) only for representation purposes. Scalebar 10 µm. (b) Dot blot of the mean red/green ratio of SH-SY5Y cells under different conditions of transfection. n > 10 for each condition; ***, p < 0.001 according to unpaired Student-t test with unequal variance. (C) FACS flowcytometry confirms the inhibitory effect of the P1 mutation on α-secretase, but not β-secretase, processing. n = 3 independent experiments; error bars, s.d.; *, p < 0.05, ***, p < 0.001, according to unpaired Student-t test with unequal variance. Scatterplots from one experiment are reported in Fig. S7.

The P1 mutation inhibits the processing of APP by ADAM10

SH-SY5Y neuroblastoma cells were co-transfected with mChAPPmGFP or mChAPPP1mGFP and ADAM10mBFP in order to test whether the P1 mutation really inhibits the α-secretase activity. We compared the mean red/green ratio in cells overexpressing ADAM10mBFP and mChAPPmGFP with or without the P1 mutation (Figs. 7A–7B). The red/green ratio was found to be significantly higher in cells overexpressing mChAPPP1mGFP compared to mChAPPmGFP, showing not only that the ADAM10mBFP construct is proteolytically active, but also that the presence of the P1 mutation strongly prevents the cleavage by α-secretase. These results were validated by FACS flow cytometry (Fig. 7C and Fig. S7). SH-SY5Y cells were co-transfected with mChAPPmGFP or mChAPPP1mGFP and ADAM10mBFP or mBFP. As above, two different areas containing the same number of cells co-transfected with mChAPPmGFP and mBFP were used to gate the scatter plots of cells co-transfected with the other conditions and calculate the mean ratio between the number of cells in the red area over the number of cells in the green area. The number of cells in the red area significantly increased with the overexpression of mChAPPP1mGFP compared to the wild-type, both in the presence of ADAM10mBFP or mBFP, denoting an almost complete resistance to α-secretase proteolysis. Moreover, SH-SY5Y cells were co-transfected with mChAPPP1mGFP and BACE1mBFP in order to verify potential side-effects of the P1 mutation on the β-secretase processing. The ratio between the number of cells in the red and green areas resulting from the co-expression of mChAPPP1mGFP and BACE1mBFP is comparable with that from mChAPPmGFP and BACE1mBFP. This data suggests that the variant of APP carrying the P1 mutation is hardly processed by ADAM10, while still cleavable by Bace1.

Discussion

We have successfully demonstrated the proof of principle of a bioassay to evaluate the processing of APP directly in single living cells, a less biased method compared to the in vitro kits that are commercially available. To estimate the rate of processing of APP, we measured the variation in the red/green fluorescence ratio emitted from cells overexpressing mChAPPmGFP. We demonstrated that mChAPPmGFP is correctly folded and targeted to the plasma membrane, and it is still cleavable by the β-secretase Bace1. We have noticed that a similar approach has been tested in two previous works (Coughlan et al., 2013; Villegas, Muresan & Ladescu Muresan, 2014). In one of these papers, the residual C-terminal region of APP770 (106 amino acids) carrying the Swedish mutation was fused with two different fluorescent proteins (DsRed2 and EGFP) at the opposite terminal sides (Coughlan et al., 2013). By contrast, our mChAPPmGFP construct is generated from the full length APP695 and contains the original signal-peptide sequence that is required for targeting the newly synthesized protein to the plasma membrane. Furthermore, in order to avoid oligomerization events, we used two monomeric fluorescent proteins, mCherry and mGFP. These two fluorescent proteins have comparable maturation times of approximately 10–15 min (Shaner et al., 2004; Subach et al., 2008), thus limiting the rise of artifacts due to different maturation events. In agreement with the fact that the mCherry and the mGFP tags are kept more than 10 nm apart by the plasma membrane, we did not observe any Foster resonance energy transfer (FRET), preserving therefore the linearity of the red/green ratio with respect of the amount of cleaved APP (Fig. S8). In the other interesting work (Villegas, Muresan & Ladescu Muresan, 2014), APP695 was fused to CFP and YFP at the two opposite terminals, with CFP correctly positioned after the signal peptide, to study the localization of APP fragments and the trafficking processes occurring intracellularly. The authors found that the presence of the tags did not alter the processing of APP by the secretases, and that some intracellular N-terminal fragments—possibly corresponding to our observed red punctae—do not colocalize with full length APP or C-terminal fragments, in complete agreement with our data. Importantly, the dual tagged APP, at low level of expression, showed a subcellular distribution and a proteolytical profile highly comparable to that of endogenous APP.

Interestingly, our results show that cells of the same type (both neuronal and neuroblastoma cells) display different ratios of red/green fluorescence intensity, suggesting that the processing rate of APP is highly variable even within the same population. Furthermore, a certain level of heterogeneity in the red/green intensity ratio is also found within the same cell, indicating that the activity of the proteases can vary in different districts of the cell. This finding is in agreement with previous results showing that there are differences in the processing of APP depending on its targeting on the plasma membrane of highly polarized cells (Haass et al., 2012; DeBoer et al., 2014).

It has been shown that APP can be internalized from the cell surface and specifically targeted to lysosomes (Tang et al., 2015). Previous works demonstrated that the fluorescence emitted by GFP can be strongly reduced by the acidic pH of the lysosomal lumen, while the fluorescence emitted by mCherry remains stable in a broad range of pH (Katayama et al., 2008; Mizushima, Yoshimori & Levine, 2010). These considerations could in part explain the presence in our images of intracellular vesicles with high red fluorescence intensity barely co-localizing with the green fluorescence signal. However, the fact that we did not observe any co-localization with the lysosome marker LAMP1 does not convincingly support this hypothesis. It is rather plausible that the red punctae correspond to cleaved N-terminal fragments transported in vesicles separated from those carrying the full length protein and the C-terminal fragments, as already observed by Villegas, Muresan & Ladescu Muresan (2014).

As an alternative to the siRNAs, we have created a variant of mChAPPmGFP carrying a mutation capable of inhibiting the α-secretase processing without altering the β-secretase cleavage. The importance of this construct is the possibility to create transgenic animals, like zebra-fish and rat, which could be used to produce a 3D map of the whole brain showing areas with potentially different β-secretase activities. Previous data (Sisodia, 1992), together with our confocal microscopy and FACS results, suggest that the P1 mutation makes APP cleavage-resistant by α-secretase. However, it is unclear whether it completely inhibits the APP processing. Although cells co-transfected with ADAM10mBFP and mChAPPP1mGFP have a cell number in the red/green area ratio similar to that of cells co-expressing mBFP and mChAPPmGFP, the ratio for mBFP and mChAPPP1mGFP transfected cells is much higher than the ratio for ADAM10mBFP and mChAPPP1mGFP transfected cells. It is possible that under conditions where ADAM10 is strongly overexpressed, a small fraction of mChAPPP1mGFP can be still cleaved.

In general, FACS flow cytometry corroborated the reliability of our bioassay and confirmed the heterogeneous processing of APP within the same type of cells. Furthermore, the possibility to separate cells with different rates of APP processing, i.e., different red/green ratios, using the cell-sorting capability of FACS might allow to perform a complete proteomic, lipidomic and transcriptomic analysis of these sub-populations of cells, and to address one of the most puzzling questions in the context of Alzheimer’s disease: why the proteolytic processing of APP is higher in certain cells than in others.

The combined approach presented in this work could be also used as versatile bioassay to test in living cells the effect of specific molecules on the proteolysis of APP. A more detailed knowledge of the factors governing the processing of APP could contribute to develop alternative drugs for AD treatments.

Supplemental Information

Supplemental Information 1 Supplementary Information

Click here for additional data file.

We thank Dr Francesco Bemporad, Dr Daniele Nosi, Dr Franco Quercioli and Rachele Reggioli for their technical help.

Additional Information and Declarations

Competing Interests

Author Contributions

Data Availability

The authors declare there are no competing interests.

Niccoló Parenti and Martino Calamai conceived and designed the experiments, performed the experiments, analyzed the data, wrote the paper, prepared figures and/or tables.

Ambra Del Grosso performed the experiments, reviewed drafts of the paper.

Claudia Antoni performed the experiments, analyzed the data, wrote the paper, prepared figures and/or tables.

Marco Cecchini and Renato Corradetti contributed reagents/materials/analysis tools, reviewed drafts of the paper.

Francesco S. Pavone contributed reagents/materials/analysis tools.

The following information was supplied regarding data availability:

Repository: Figshare.

Fluorescence spectrometer raw data:

Calamai, Martino (2017): Fluorescence spectrometer raw data. Figshare.

https://doi.org/10.6084/m9.figshare.3841308.v1.

FACS raw data:

Calamai, Martino (2017): FACS raw data Fig. 6. Figshare.

https://doi.org/10.6084/m9.figshare.3841302.v1.

Calamai, Martino (2017): FACS raw data Fig. 7. Figshare.

https://doi.org/10.6084/m9.figshare.4658779.v1.

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
