# Peer review of "Direct imaging of APP proteolysis in living cells"

_PeerJ, doi:10.7717/peerj.3086_

## Round 0.1 · original submission · Major Revisions

· Academic Editor

Major Revisions

Two experts reviewed your paper and found it is valuable. But they raised some problems on the current version of the manuscript. I hope you find their comments valuable and revise according to their comments.

Reviewer 1 ·

Basic reporting

The language used by the authors to describe the methodology is unambiguous and pertinent, the subject is well contextualized and the figures reflect what is described in the text. Further, Parenti et al. use several experimental approaches to validate their method.

Experimental design

In this study, Parenti et al propose and validate a new in vivo method for measuring amyloid precursor protein (APP) processing as an alternative to commercially available kits that measure secretase activity in vitro. This new method is a rapid bioassay that would generate direct insight into the proteolytic cleavage of APP directly in cells. Further, Parenti et al. use several experimental approaches to validate their method. This new method has the potential to be an important tool in APP research because it allows us to spatially visualize APP processing in a live-cell system.

Validity of the findings

Overall, the experimental design is appropriate and the data are reliable. The findings are of considerable interest. However, there are several points that require attention.

- In the introduction, the authors mention that the Aβ oligomers contribute “mostly” to the development of the disease. This is a controversial issue on the field, oligomers have an undeniable role in the toxicity process but there are many other factors that contribute to the development of the disease in earlier stages.

- The authors use the word “membrane” synonymously with “plasma membrane.” There are many other membranes in the cell. I suggest that the authors kindly indicate which one they are referring to in order to avoid confusion.

- In Figure 2a, the authors show that anti-mCherry immunocytochemistry (blue) doesn’t match with mCherry (red). The authors say that the method used is “surface labeling” but in the methods they say that the cells were permeabilized. If the cells were permeabilized, how is the protein detection discrepancy possible?

- To verify the silencing the authors show, in supplemental figure 2b, an image of Bace1-mBFP. I strongly recommend to perform a western blot anti Bace1 in order to confirm this result.

- The authors observed small red fluorescent punctae that don’t correspond to a comparable level of green fluorescence intensity. They suggest that these “red punctae” localize in lysosomes but they didn’t add any subcellular marker to confirm this hypothesis. Have they considered the possibility of localization in endosomes? Mature active forms of Bace1 are largely localized in late Golgi and early endosome compartments (Vassar et al., J Neurosci, 2009). I strongly recommend adding subcellular markers for both endosomes and lysosomes to confirm the subcellular localization of mChAPPmGFP and its cleavage products.

- The authors state that mostly APP localizes in the Golgi (an appropriate reference for this is needed). A Golgi marker in the immunocytochemistry is necessary to confirm this claim.

- In the western blot shown in Figure 4d, I would expect to see the C-terminal fragments of mChAPPmGFP. These fragments should be present in the pellet samples when Bace1 is not silenced when blotting for antiGFP and antiAβ. Why are these fragments not shown? The analysis of the C terminal fragments in the gel could be useful to corroborate the appropriate processing of mChAPPmGFP.

- To see if mChAPPmGFP can also be processed by γ-secretase, I would propose to add a γ-secretase inhibitor (DAPT) to see if the fluorescent protein accumulations are Aβ peptides or CTFs.

- The authors use the silencing of Bace1 as a way to discrimate between α- and the β-cleavage. Since they are overexpressing Bace1, they assume that mCherry-sAPPα levels would be negligible. However, since they detect mCherry-sAPPα when silencing Bace1, I would consider it appropriate to add control conditions that consist of treatment with an α-secretase inhibitor or silencing of α-secretase to completely abrogate mCherry-sAPPα production.

Reviewer 2 ·

Basic reporting

Minor comments. repeated "the" at end of abstract, "length" misspelled (line 115). Sentence structure was awkward throughout. Consider having more people edit this paper.

Experimental design

Consider using HEK cells for FACS analysis. Easily transfected, no trypsin required to detach. You are almost certainly losing some APP to trypsin proteolysis at the cell surface.

Validity of the findings

Note that Villegas et al. (Hum Mol Genet. 2014 Mar 15;23(6):1631-43) made a similar dual tagged APP construct as you did, and found that the N and C termini are trafficked differently. Please cite this paper and discuss, because I think it has implications on the effectiveness of your method. If N terminal fragments of APP are retained intracellularly (as C terminal fragments are), this method's power to detect APP cleavage would be limited.

Additional comments

I think your design and methods are generally correct. Given the amyloid hypothesis, a construct that would selectively detect the beta cleavage of APP would be the most compelling. As such, you demonstrate that your method can detect beta cleavage, however, it would also detect alpha cleavage as well, as you also demonstrate. If you could design a tag that would could be selectively detected when present in sAPPbeta and not sAPPalpha, that would be an excellent tool for AD.

---

## Round 0.2 · accepted · Accept

· Academic Editor

Accept

Authors clarified all the concerns raised by the reviewers. Therefore, this paper is now accepted for publication.